# Like Mother, like Son: Physical Activity, Commuting, and Associated Demographic Factors

**Caroline Brand [1], Cézane Priscila Reuter [1], Arieli Fernandes Dias [2], Jorge Mota [3], Michael Duncan [4], Anelise Reis Gaya [2], Luiza Naujorks Reis [2], Jane Dagmar Pollo Renner [1] and Emilio Villa-González [5,*]**

1 Graduate Program in Health Promotion, University of Santa Cruz do Sul –UNISC. 2293 Independência Av, Santa Cruz do Sul 96815-900, Brazil; carolbrand@hotmail.com.br (C.B.); cezanereuter@unisc.br (C.P.R.); janerenner@unisc.br (J.D.P.R.)
2 School of Physical Education, Physiotherapy and Dance, Graduate Program in Human Movement Sciences, Federal University of Rio Grande do Sul, 750, Felizardo Street, Porto Alegre 90690-200, Brazil; ariieli_dias@hotmail.com (A.F.D.); anegaya@gmail.com (A.R.G.); luiza_n_reis@hotmail.com (L.N.R.)
3 Research Center on Physical Activity, Health and Leisure (CIAFEL). Faculty of Sport, University of Porto. 91, Dr. Plácido da Costa Street, 4200-450 Porto, Portugal; jmota@fade.up.pt
4 Centre for Sport, Exercise and Life Sciences, Alison Gingell Building, Coventry University, Coventry CV1 5FB, UK; aa8396@coventry.ac.uk
5 Department of Physical and Sports Education, PROFITH "PROmoting FITness and Health through Physical Activity" Research Group, Sport and Health University Research Institute (iMUDS), Faculty of Education and Sport Sciences, University of Granada, 52005 Melilla, Spain
* Correspondence: evilla@ugr.es; Tel.: +34-637942790; Fax: +34-958-24-43-69

**Abstract:** A mother's healthy conduct may lead to the healthy conduct of their children. Thus, this study aimed to verify the role of demographic factors in the relationship between mothers' physical activity (PA) and commuting to work with children and adolescent's PA and commuting to school. This cross-sectional study comprised 1421 children and adolescents aged 6 to 17 years and 1421 mothers, from Brazil. PA, commuting, socioeconomic status (SES), skin color/ethnicity, and living area were evaluated by questionnaire. Logistic binary regression models were used. Results indicated that mothers' PA and commuting were associated with children and adolescent's PA and commuting to school in crude and adjusted models. Considering the role of the demographic factors, an association was only observed for girls in the relationship between mother's PA with children's PA. In adolescents, an association was observed in both high/low SES, boys/girls, and rural/urban areas. Regarding children and adolescent active commuting to school, there was an association with mothers commuting. All demographic factors were strongly associated, except for rural areas. Therefore, mothers' PA as well as commuting to work are associated with children and adolescent's PA and commuting to school. Sex, living area, and SES are the related demographic factors.

**Keywords:** parent-child relations; socioeconomic factors; transportation

---

## 1. Introduction

Globally, physical activity (PA) levels among children and adolescents are low [1]. Although 81% of students are classified as insufficiently active, the prevalence of inactivity decreased in boys between 2001 and 2016 (from 80.1% to 77.6%), while there was no significant change for girls (from 85.1% to 84.7%) [1]. This is a worrying scenario, as regular PA practice is related to several benefits in the pediatric population. This includes cardiometabolic and bone health [2,3], as well as positive influences on cognitive functions [4] and symptoms of depression and anxiety [5]. In addition, PA engagement

during childhood is associated with positive health outcomes later in life [6]. It is therefore important to enable children to develop positive PA habits for both current and future health.

There are different domains related to PA, such as transport, occupation, recreation, and household [7]. Active commuting is defined as walking or cycling to and from school/work for an indeterminate period [8]. Among youngsters, active commuting to school (ACS) (walking or cycling) has been proposed as an optimal strategy to enhance PA levels, increasing the chance to achieve PA recommendations by 42% in boys and 66% in girls [9]. Likewise, children that walk to/from school showed approximately 6 more minutes of moderate to vigorous PA (MVPA) per day, compared to their passive commuter peers [10]. Furthermore, a recent longitudinal Spanish study indicated that ACS may help to attenuate declines in MVPA that are usually associated with growth [11]. Moreover, this active behavior is associated with a reduction in glucose and low-density lipoprotein cholesterol levels, with better physical fitness and lower incidence of metabolic syndrome in children and adolescents, respectively [12,13].

In youths, several factors have been described as determinants for an active lifestyle, including PA and commuting to school. In the present study, we highlight the role of family environment, since it is the context in which the child begins to develop and may be a determinant for their lifestyle choices. Thus, parents are the behavioral models for cognitive, physical, motor, and affective development of their children [14,15]. In addition, the parents' view of education, interpersonal relationships, and parenting styles in general are determinant factors in the optimal development of the children. Previous studies have investigated the influence of a mother and father's behaviors on children's PA and commuting to school and found controversial results. In a previous Portuguese study, it was observed that both the mother and father had a similar influence on their offspring's PA levels [16]. A recent systematic review indicated a weak association between parent and child PA [17]. However, a study carried out in Belgium, Greece, Hungary, Germany, and Norway, showed that maternal but not paternal participation in sport, outdoor activities, and active commuting were associated with children's higher participation in those activities [18]. This association between mothers' PA, but not fathers' PA, and children's PA was similarly identified by Jacobi et al. [19] using a French sample. In general, and specifically in the context of Brazil, women are mainly responsible for childcare and consequently spend more hours with children compared to male [20]. Thus, it is hypothesized that mothers could be more determinant in children's behavior development. Therefore, in the present study, the mother's role in their children PA will be specifically addressed.

It is also worthy of note that some investigations have addressed the role of sociodemographic factors in both active behaviors, i.e., PA and ACS [21,22], that are also examined in the current study. However, to the authors' knowledge, there are no studies that have investigated how sex, socioeconomic status, and living area could be of influence in the association between a mother's lifestyle with children's lifestyle (concerning PA and ACS). Therefore, the present study aimed to verify the role of demographic factors in the relationship between mother's PA and commuting to work with children and adolescent's PA and commuting to school.

## 2. Materials and Methods

### 2.1. Design and Sample

A cross-sectional study comprising 1421 children (mean age 9.13 ± 1.45) and adolescents (mean age 13.68 ± 1.50), aged between 6 and 17 years, and their mothers (n = 1421) was carried out. Students were selected by conglomerate from 19 public and private schools in Santa Cruz do Sul, Rio Grande do Sul, Brazil. In 2004, a survey was conducted in this city that indicated the number of schools (n = 50) and students (n = 17,688) enrolled. A sample size calculation was made considering the population density of schoolchildren in all regions of the city (south, north, east, west, center) including public (municipal and state) and private schools. Schools were subsequently randomly selected and invited to form a cohort. The schoolchildren's mothers or legal guardians signed free and informed consent

forms, in which it was mentioned that they should answer a questionnaire, and also indicating that they authorize the participation of their children. This study was approved by the Human Research Ethics Committee of the University of Santa Cruz do Sul (UNISC) under certificate number 1.498.305.

## 2.2. Instruments and Procedures

Data was collected in 2016 and 2017 at the university. All variables considered in the present study were self-reported. To obtain the information regarding physical activity and active commuting, the questionnaire "Lifestyle, Health and Welfare - child/adolescent" by Barros and Nahas [23] was adapted by the researchers to meet the objectives of the project. Mothers answered the following questions. (1) Do you currently practice any sport/physical activity? (2) How do you predominantly go to work? Children answered the following questions. (1) Do you currently practice any sport/physical activity? (2) How do you predominantly go to school? For questions 1, the answers options were yes or no, while for questions 2 the options were bus, on foot, car or motorcycle, bike, other (please specify). Answers were classified into active (walking, cycling, or other active forms of transport requiring a physical effort) or passive means of transportation (including alternatives such as bus, car, motorcycle, school bus, and other types of passive motor vehicle).

The socioeconomic status was classified according to Brazilian Association of Research Companies (ABEP) [24]. This classification considers the head of the household's educational level and the number of certain items they have, such as car, washing machines, bathrooms, among others. Each answer was scored and the sum of these scores was used as a measure of each family's social class. Participants were subsequently classified into eight distinct economic classes (A1, A2, B1, B2, C1, C2, D, and E). For the present study, these classes were regrouped in two: high economic classes (A1, A2, B1, and B2), and low economic classes (C1, C2, D, and E). Participants under the age of eleven were considered children and those aged twelve or more were classified as adolescents [25].

Tanner's criteria [26] were used to determine sexual maturation, using images of the development of breast stages for girls, genital stages for boys, and pubic hair for both. Children and adolescents were asked to indicate the image according to their current stage. Five sexual maturation stages were considered and then categorized into prepubertal (stage I), initial development (stage II), continuous maturation (stages III and IV), and matured (stage V).

Skin color/ethnicity was also reported by the participants, according to the following options: white, black, brown/mulatto, indigenous, and yellow. Area of residence was reported, i.e., adolescents were asked to indicate if they lived in rural or urban.

## 2.3. Statistic Analysis

Descriptive procedures were performed for all variables, and results were presented as absolute and relative values. All variables were checked for normality through the Shapiro-Wilk test, then differences between children and adolescents were determined through the chi-square test. To verify internal consistency from the questions regarding physical activity and commuting, Cronbach Alpha was calculated (0.25). Different logistic binary regression models were fitted for children and adolescents as follows: association between mother's PA and children's PA (crude-model 1); association between mother's PA and children's PA adjusted for sex, age, area, socioeconomic status, color/ethnicity, and maturational stage (model 2). The other models considered the association between mother's PA and children's PA (model 2), stratified by sex (model 3), living area (model 4), and socioeconomic status (model 5). The same procedures were used to test the association between mother's active commuting to work and children's ACS. All analyses were carried out using the IBM SPSS 22 (SPSS, Inc., Chicago, IL, USA), alpha <0.05 was adopted, and confidence intervals (95%) and Nagelkerke $R^2$ values were presented.

Logistic binary regression was used as a statistical test for sample calculation on G* Power 3.1 program (Heinrich-Heine-Universität-Düsseldorf, Düsseldorf, Germany), with test power (1-β) = 0.95,

a significance level of $\alpha = 0.05$, and effect size of 0.03. The number of predictors considered was six and the minimum sample size needed was established as 702 in each age group (children and adolescents).

## 3. Results

Descriptive data are shown in Table 1. The majority of the sample was composed of girls (56.8%). In addition, most of the children (62.7%) and adolescents (57.2%) were classified in the low socioeconomic status. Regarding the lifestyle habits, 72% of mothers (for children) and 64.5% (for adolescents) reported that they do not practice PA and sports, while 75.7% of mothers (for children) and 70.4% (for adolescents) usually use passive commuting to work. Children and adolescents, 43.9% and 37.6%, respectively, do not practice PA and sports, and 58.6% and 50.2% use passive commuting to school.

**Table 1.** Descriptive characteristics stratified by children and adolescents.

| Variables | Children (n= 708) n (%) | Adolescents (n= 713) n (%) | Total (n= 1421) n (%) |
|---|---|---|---|
| Sex | | | |
| Boys | 309 (43.6) | 305 (42.8) | 614 (43.2) |
| Girls | 399 (56.4) | 408 (57.2) | 807 (56.8) |
| Maturational stage | | | |
| Pre-pubertal | 297 (41.9) | 15 (2.1) | 312 (22.0) |
| Initial development | 247 (34.9) | 92 (12.9) | 339 (23.9) |
| Continuous maturation (stage III-IV) | 143 (20.2) | 497 (69.7) | 640 (45.0) |
| Maturated | 21 (3.0) | 109 (15.3) | 130 (9.1) |
| Skin color/ethnicity | | | |
| White | 609 (86.0) | 571 (80.1) | 1180 (83.0) |
| Black | 32 (4.5) | 40 (5.6) | 72 (5.1) |
| Brown/mulatto | 66 (9.3) | 93 (13.0) | 159 (11.2) |
| Indigenous | 0 (0.0) | 5 (0.7) | 5 (0.4) |
| Yellow | 1 (0.1) | 4 (0.6) | 5 (0.4) |
| Living area | | | |
| Rural | 89 (12.6) | 107 (15.0) | 196 (13.8) |
| Urban | 619 (87.4) | 606 (85.0) | 1225 (86.2) |
| Socioeconomic status * | | | |
| High | 264 (37.3) | 305 (42.8) | 569 (40.0) |
| Low | 444 (62.7) | 408 (57.2) | 852 (60.0) |
| Mother's physical activity * | | | |
| Yes | 198 (28.0) | 247 (34.6) | 445 (31.3) |
| No | 510 (72.0) | 466 (65.4) | 976 (68.7) |
| Mother's commuting to work * | | | |
| Active | 172 (24.3) | 211 (29.6) | 383 (27.0) |
| Passive | 536 (75.7) | 502 (70.4) | 1038 (73.0) |
| Children's physical activity * | | | |
| Yes | 397 (56.1) | 445 (62.4) | 842 (59.3) |
| No | 311 (43.9) | 268 (37.6) | 579 (40.7) |
| Commuting to school * | | | |
| Active | 293 (41.4) | 355 (49.8) | 648 (45.6) |
| Passive | 414 (58.6) | 358 (50.2) | 773 (54.4) |

* Statistically significant difference in Chi-square test $p < 0.05$.

Figure 1 highlights an association between mother's PA with children's and adolescent's PA. For children, an association was found in the crude model and also in the adjusted model. When the role of demographic factors was considered, an association was observed but only for girls. In adolescents, the associations were found in all models; thus, mothers that are physically active increase the odds

ratio of the adolescent's PA practice compared to the inactive mothers, considering that living area, sex, and socioeconomic status were the demographic factors related.

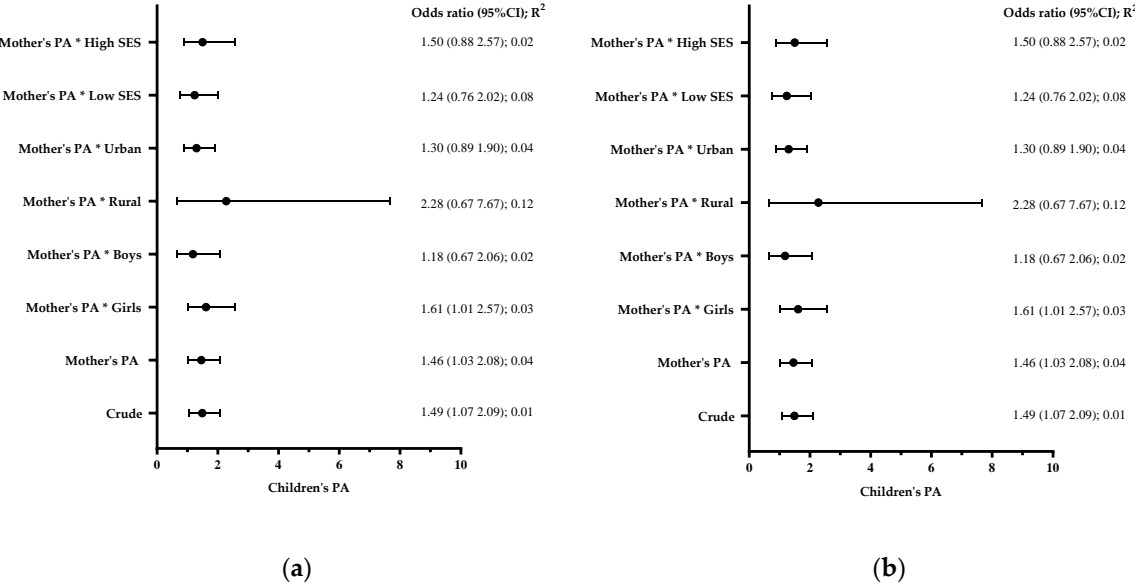

(a)    (b)

**Figure 1.** Association between mother's physical activity with children's (**a**) and adolescent's (**b**) physical activity stratified by sex, area, and socioeconomic status. PA: Physical activity. SES: Socioeconomic status. All models were adjusted for demographic factors additionally to color/ethnicity and maturational stage, except for the crude model. Values on the x-axis are odds ratio (95% confidence interval).

Regarding children's ACS, an association was found with the mother's active commuting to work and this association remained statistically significant when stratified by sex, urban area, and socioeconomic status (Figure 2a). The same results were found for adolescents (Figure 2b).

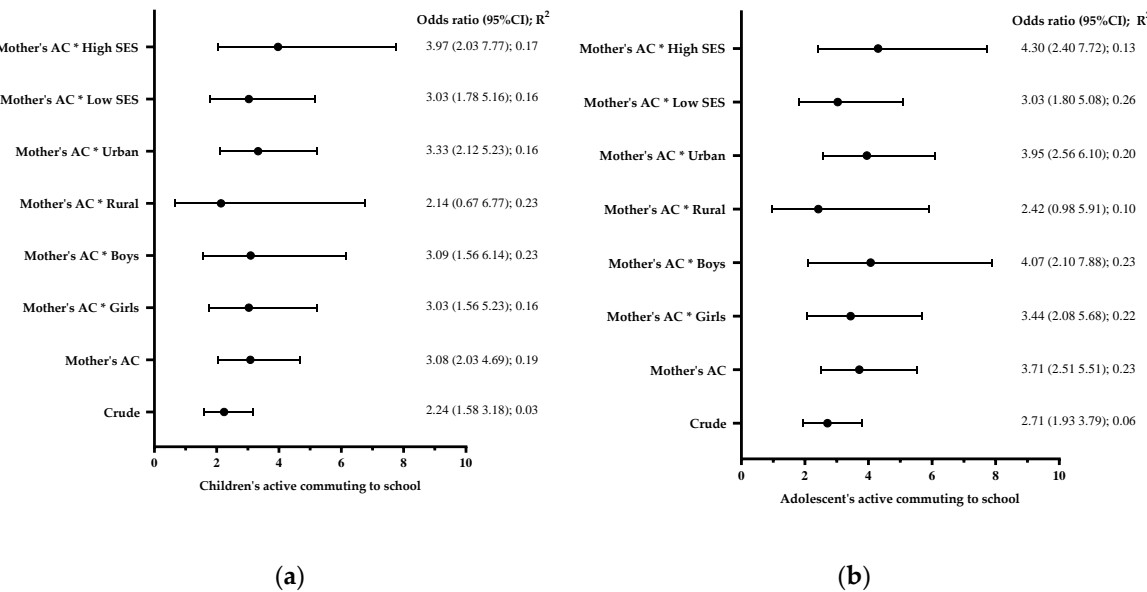

(a)    (b)

**Figure 2.** Association between mother's active commuting to work with children's (**a**) and adolescent's (**b**) active commuting to school stratified by sex, area, and socioeconomic status. AC: Active commuting. SES: Socioeconomic status. All models were adjusted for demographic factors additionally to color/ethnicity and maturational stage, except for the crude model. Values on the x-axis are odds ratio (95% confidence interval).

## 4. Discussion

The main findings of the present study indicate that a mother's PA is associated with PA undertaken by their children. For children, this association was only observed in girls, whereas sex, area (rural vs urban), and socioeconomic status influence this relationship for adolescents. For active commuting for both children and adolescents, the present study suggests that the mother's commuting mode influence and all demographic factors were strongly related, except for living in rural areas. Therefore, the results of the current study suggest a mother's PA behaviors play an important role in their son's PA behaviors.

Childhood is a critical phase for the development of a healthy lifestyle throughout life. Almost all of the opportunities and choices made in this childhood period are influenced by the family environment so that parental habits, choices, and perceptions play a determining role in a child's routine [15,27,28]. For example, children of parents who offer more autonomy for commuting unaccompanied and actively to school have a better understanding of safety issues [27]. Parental perception of barriers, such as volume and speed of traffic, crime, and distance from home to school are related factors of ACS [15]. In addition, access, time, and costs are barriers for PA practice [28]. It is well known that lifestyle characteristics of parents are closely related to those developed by their children. In the current study, considering the specifics of the context in Brazil, we specifically considered the role of the mother as females in Brazil work on average 4.8 hours less per week than males [29] and spend more time in childcare and housework [20].

In our sample, approximately 31% of mothers reported being physically active, while 56.1% of children and 65% of adolescents reported practicing sport or PA. Globally, it is estimated that only 19% of students aged 11–17 years achieve physical activity recommendations [1]. A higher prevalence of PA was observed in mothers from Switzerland, once 46.6% reported practicing exercise often [30]. A study conducted with Brazilian mothers and their children indicated that approximately five out of ten adolescents practiced 300 min or more per week of moderate to vigorous PA (51.1%), while for mother the prevalence was about 14% [31].

Our results identified an association between mother's PA and their children's PA. Thus, our data seems to highlight the potential role of mothers in their children's PA behavior. One study undertaken with Brazilian adolescents has indicated that PA of the boys and girls was correlated with their father and mother's PA [32]. In Portuguese children, prior research has reported that girls with physically active mothers were engaged in more sports and practiced sport 1–2 times per week [33]. However, the results of the aforementioned studies have only considered sociodemographic factors as adjustment variables. The findings presented in the current study also parallel those of other studies that used accelerometers to objectively measure PA. Yoon et al. [34] and Jago et al. [35] found that children's moderate to vigorous PA levels were significantly associated with their mother's vigorous PA levels. Despite this, a comparison of the results of the current study with those reported previously should be made with caution as the current study assessed PA using questionnaires, and the prior aforementioned work employed accelerometry.

It is noteworthy that in children, the association between mother's PA and children's PA was found only in girls (OR = 1.61; 95% CI = 1.01 2.57). A possible explanation for this could be that daughters usually have a closer relationship with their mothers than do boys, especially during childhood, which is in accordance with the study by Yoon et al. [34]; however, this study has found this correlation only in children, while our study observed that mother's PA was related also with adolescent's PA, in both boys and girls. Indeed, our results indicate that sex, socioeconomic status, and living in an urban area are related factors for adolescents. These age group differences could be explained because children have less responsibility for making decisions about lifestyle factors such as PA and commuting, thus maternal influence may diminish when children get older and become more independent [36]. Other aspects could be also relevant, such as socioeconomic status and living area.

Regarding ACS, a study conducted in the United States indicated that only 16.5% of youth walk to or from school at least once a week [37]. In Ecuadorian school-aged students, the prevalence of

ACS was 21.5% [38], while in Mexican children and adolescents this prevalence was much higher at 62.2% [39]. Similarly, 62.4% of Spanish children and adolescents actively commute to school [40]. More recently, it was observed that 56.1% of adolescents from 63 low- and middle-income countries from different regions (Asia, Africa, Latin America, and Oceania) actively commute to school [9]. Finally, in Brazilian cities, the prevalence of ACS was 41% in children from Florianopolis [41] and 63% in adolescents from São Paulo [42], while in the present study, the findings for children were similar (41.4% and 49.8% for adolescents). In Porto Alegre, the prevalence of ACS in adolescents was lower (30.2%) [43]. This would, therefore, suggest that there is great variation in this behavior when considering different locations, which may be explained by a variety of factors related to active commuting, such as neighborhood environment, psychosocial factors, and family context [43,44].

The influence of parental behavior, especially concerning active commuting to work has also been previously explored. Indeed, our findings disclosed that the mother's active commuting to work is associated with children and adolescent's ACS. Also, the analysis of demographic factors showed that this association was observed for both sexes, high and low socioeconomic status, and urban area. In accordance with our findings, other evidence has reported that children and adolescents whose parents usually active commute to work used to engage in higher ACS [40,45–47].

However, none of the aforementioned studies investigated the role of demographic factors in the association between parents' active commuting to work and commuting to school. The focus on this aspect is one of the novel contributions of the current study. We highlight that this is a complex issue once this relationship is ascribed to lifestyle aspects involving the interactions of mothers and their children and that it is essential to understand the influence of intervenient variables. Living in a rural area was the only demographic factor that was not related to ACS in children and adolescents. This may be due to the fact that the majority of the sample derived from urban areas (86.2%), and, indeed, it seems that the main facilitating factor for active commuting is living in an urban area, where the distances are shorter and there is a greater probability of active commuting [48]. Moreover, many students living in rural areas need to commute to a nearby city for study and, therefore, use the passive mode of transportation.

Generally, a mother's healthy conduct may lead to the healthy conduct of their children. Indeed, mothers seem to participate more actively in the children's routine, as they usually complete parent surveys (as observed in the present study) and participate in family-based interventions [49]. Similarly, some studies that considered both maternal and paternal influence, showed that the magnitude of the correlation coefficients was higher for mothers compared to fathers [18,19,30].

Therefore, active commuting to school and physical activities are multifactorial behaviors and different associated factors should be investigated, including the role of the mother's lifestyle and all the demographic factors that may influence. Thus, as practical implications, we highlight that interventions targeting increasing PA and active commuting among children and adolescents should also consider the relevance of the influence of mothers on determining healthy habits.

There are some limitations to the present study, which needs to be considered. First, the evaluation of the PA of both mothers and adolescents was self-reported, which could lead to an overestimation of time spent in PA. The self-report method employed also prevented the determination of PA intensity. Second, we only have reported information on the physical activity and commuting of the mother, thus we could not compare maternal and paternal influence. Third, we could not determine the role of the distance from home/work in commuting as we did not evaluate this measure. Fourth, the study is limited by the cross-sectional nature of the data, preventing conclusions from being drawn about the direction of the association. Despite this, there are some important strengths of the current study that must be acknowledged, such as the relatively large sample size of Brazilian children and adolescents. Moreover, although some studies have investigated the relationship between mother's PA and commuting with children and adolescent's PA and commuting to school, they failed to consider the influence of sociodemographic factors in this association. The current study addresses this gap.

## 5. Conclusions

The mother's PA and commuting to work are associated with children's PA and commuting to school, and sex, living area, and socioeconomic status are the related demographic factors. Therefore, our data suggest that interventions focused on changing PA and commuting behaviors should target both mothers and youth to increase their effectiveness.

**Author Contributions:** Conceptualization, C.B., A.F.D., and E.V.-G.; methodology, C.P.R. and J.D.P.R.; software, C.B. and A.F.D; validation, C.P.R., J.D.P.R., and A.R.G; formal analysis, C.B. and A.F.D.; investigation, C.P.R. and J.D.P.R.; resources, C.P.R. and J.D.P.R.; data curation, C.B., A.F.D., and L.N.R.; writing—original draft preparation, C.B., A.F.D., L.N.R., and E.V.-G.; writing—review and editing, A.R.G., J.M., and M.D.; visualization, A.R.G., J.M., and M.D.; supervision, E.V.-G., J.D.P.R., and C.P.R.; project administration, C.P.R. and J.D.P.R. funding acquisition, C.B., and L.N.R. All authors have read and agreed to the published version of the manuscript.

**Funding:** This research was funded by the Coordination for the Improvement of Higher Education Personnel (CAPES) and Foundation for Science and Technology (FCT), Portugal (SFRH/BSAB/142983/2018 and UID/DTP/00617/201).

**Acknowledgments:** We would like to express our gratitude to the children, parents, and schools participating in the study.

**Conflicts of Interest:** The authors declare no conflict of interest.

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
