# Peer review of "Like Mother, like Son: Physical Activity, Commuting, and Associated Demographic Factors"

_sustainability, doi:10.3390/su12145631_

Round 1

Reviewer 1 Report

See attach

Author Response

Dear Editor,

Please find enclosed the revised Manuscript number 859530 entitled Like mother, like son: physical activity, commuting, and associated demographic factors” which we submitted to the “Sustainability”.  The comments to the reviewer(s) are included at the bottom of this letter.

We would like to acknowledge the reviewers’ comments to the manuscript, and we hope allowed us to improve the quality of the work presented. We have carefully considered all issues mentioned by the reviewers’ in this revised version of the manuscript and the alterations throughout the manuscript are highlighted in the text in yellow colour, which means that new information was included. The authors would like to clarify the raised concerns by the reviewers. Please find below a point-by-point response to the reviewers’ comments. We hope that the comments and suggestions provided by the reviewers were addressed as expected.

Best regards,

The authors.

Reviewer 1

Comments:

  1. The first question for authors is whether the mothers of children (n=1421) agreed to

participate in this study. I think is hard that mothers accept to participate, it is a pretty big number. How was done?

Authors: Thanks, we understand your concern. The mothers signed a Consent Form, in which it was mentioned that they should answer a questionnaire and also indicating that they authorize the participation of their children. The following information was included in the text: “The schoolchildren’s mothers or legal guardians signed free and informed consent forms, in which it was mentioned that they should answer a questionnaire, and also indicating that they authorize the participation of their children”. (Page 2, Lines 94-98). Finally, as it was a study evaluating lifestyle habits in young people, we believe that this fact was a good incentive for mothers to participate in the study.

  1. In this study a separate field should appear for the statistical analyzed performed.

Authors: Thank you for your suggestion, a separate field was included for -Statistic Analysis. (Page 3, Line 132).

  1. Also, for questionnaire applied what was the alpha cronbach value? The 3 questionnaire must be presented as clearly and completely as possible, it is not very well understood, please reformulate.

Authors: Done. The alpha Cronbach value was in included in the data analysis as follow: “To verify internal consistency from the questions regarding physical activity and commuting Cronbach Alpha was calculated (0.25)” (Page 3, Lines 135-137). Moreover, the questions were more clearly described, to approach the ones that were directed to mothers and children. The following was included (Page 3, lines 106-111): “Mothers answered the following questions: 1) Do you currently practice any sport / physical activity? 2) How do you predominantly go to work? Children answered the following questions: 1) Do you currently practice any sport / physical activity? 2) How do you predominantly go to school? For the questions 1, the answers options were yes or no, while for the questions 2 the options were: bus; on foot; car or motorcycle; bike; other (please specify).”

In general the article is well presented.

Thank you!

Authors: Thank you for your suggestions and for the positive feedback regarding our study.

Reviewer 2 Report

Thank you for the opportunity to review the manuscript entitled, “Like mother, like son physical activity, commuting, and associated demographic factors".

I believe this study investigated a topic relevant to the readers of “SUSTAINABILITY”. There is a growing interest in research into the determinants of health-related quality of life. Studies highlighting the importance of regular physical activity as a healthy practice and numerous investigations have shown that the practice of physical activity contributes positively to wellbeing in the population.  The role of the family in physical activity development is undeniable, with parents being the most powerful force in their children’s lives. Is very important consider the relevance influence of mothers on determining healthy habits.

This paper is well written and follows well accepted standards of academic writing. Strengths include interest, detailed analysis and importance of the study. Some weaknesses are clearly defined in limitations. However, minor revisions may prove beneficial.

The authors should consider in Materials and Methods the sections: Sample, Instruments, Design, Procedure and Data Analysis.

The introduction not analyze in detail the effects of familial socialization on children’s development. The role of the family in physical activity development is undeniable. The behaviors displayed by parents toward their children have a direct influence on children’s behavior, emotional security, and well-being. The parents’ view of education, interpersonal relationships and parenting styles are determinant factors in their children’s development.

Is need in the logistic binary regression models the values of the correct estimation of the cases, χ2, Cox & Snell’s R2 and Nagelkerke’s R2.

Finally, the instruments must appropriate to the research question. The instruments must always display two important qualities: reliability and validity.  When a factor is defined by 5 items with loadings above .50, it is a solid factor with practical relevance but the “factor” that evaluate the  physical activity and active commuting  is defined by one item.

Author Response

Reviewer 2

Comments:

Thank you for the opportunity to review the manuscript entitled, “Like mother, like son physical activity, commuting, and associated demographic factors".

I believe this study investigated a topic relevant to the readers of “SUSTAINABILITY”. There is a growing interest in research into the determinants of health-related quality of life. Studies highlighting the importance of regular physical activity as a healthy practice and numerous investigations have shown that the practice of physical activity contributes positively to wellbeing in the population.  The role of the family in physical activity development is undeniable, with parents being the most powerful force in their children’s lives. Is very important consider the relevance influence of mothers on determining healthy habits.

This paper is well written and follows well accepted standards of academic writing. Strengths include interest, detailed analysis and importance of the study. Some weaknesses are clearly defined in limitations. However, minor revisions may prove beneficial.

Authors: Thank for the positive feedback regarding our study, as well as the constructive impute.

The authors should consider in Materials and Methods the sections: Sample, Instruments, Design, Procedure and Data Analysis.

Authors: Thanks for that good point. We have included new section to clarify. Now, the material and methods section was organized according to the headings:

Page 2, Line 88: 2.1 Design and sample

Page 3, Line 102: 2.2 Instruments and procedures

Page 3, Line 132: 2.3 Statistic Analysis

The introduction not analyze in detail the effects of familial socialization on children’s development. The role of the family in physical activity development is undeniable. The behaviors displayed by parents toward their children have a direct influence on children’s behavior, emotional security, and well-being. The parents’ view of education, interpersonal relationships and parenting styles are determinant factors in their children’s development.

Authors: Thanks for your constructive comment. We totally agree with your point. Thus, we have included new information on this issue to this paragraph following your recommendations (Please, see pag. 2, line 62-68). Thanks again.

Is need in the logistic binary regression models the values of the correct estimation of the cases, χ2, Cox & Snell’s R2 and Nagelkerke’s R2.

Authors: Thank you for your suggestion, the Nagelkerke’s R2 values were presented in the figures. Also, the following was included in data analysis “All analyses were carried out using the IBM SPSS 22 (SPSS, Inc., Chicago, Illinois, USA), alpha <0.05 was adopted, confidence intervals (95%) and Nagelkerke R2 values were presented.” (Page 3, Lines 144-145).

Finally, the instruments must appropriate to the research question. The instruments must always display two important qualities: reliability and validity.  When a factor is defined by 5 items with loadings above .50, it is a solid factor with practical relevance but the “factor” that evaluate the physical activity and active commuting is defined by one item.

Authors: We understand your concern. Indeed, we recognize the limitation of using only one question to evaluate physical activity, however we consider that this question is suitable to evaluate physical activity, since is simple and easily comprehensible by children and adolescents. Concerning commuting, the question about mode of commuting to and from school has been proposed as one of the most appropriate measurements for asking about mode of commuting to school in this population (Herrador-Colmenero et al. 2014).

Herrador-Colmenero M, Perez-Garcia M, Ruiz JR, Chillon P. Assessing modes and frequency of commuting to school in youngsters: A systematic review. Pediatr Exerc Sci. 2014; 26(3):291-341. doi:10.1123/pes.2013-0120
